# A Chemist’s Perspective on the Role of Phosphorus at the Origins of Life

**DOI:** 10.3390/life7030031

**Published:** 2017-07-13

**Authors:** Christian Fernández-García, Adam J. Coggins, Matthew W. Powner

**Affiliations:** Department of Chemistry, University College London, 20 Gordon Street, London WC1H 0AJ, UK; christian.fernandez@ucl.ac.uk (C.F.-G.); a.coggins@ucl.ac.uk (A.J.C.)

**Keywords:** prebiotic chemistry, phosphate, phosphorylation, nucleotides, amino acids, general acid-base catalyst

## Abstract

The central role that phosphates play in biological systems, suggests they also played an important role in the emergence of life on Earth. In recent years, numerous important advances have been made towards understanding the influence that phosphates may have had on prebiotic chemistry, and here, we highlight two important aspects of prebiotic phosphate chemistry. Firstly, we discuss prebiotic phosphorylation reactions; we specifically contrast aqueous electrophilic phosphorylation, and aqueous nucleophilic phosphorylation strategies, with dry-state phosphorylations that are mediated by dissociative phosphoryl-transfer. Secondly, we discuss the non-structural roles that phosphates can play in prebiotic chemistry. Here, we focus on the mechanisms by which phosphate has guided prebiotic reactivity through catalysis or buffering effects, to facilitating selective transformations in neutral water. Several prebiotic routes towards the synthesis of nucleotides, amino acids, and core metabolites, that have been facilitated or controlled by phosphate acting as a general acid–base catalyst, pH buffer, or a chemical buffer, are outlined. These facile and subtle mechanisms for incorporation and exploitation of phosphates to orchestrate selective, robust prebiotic chemistry, coupled with the central and universally conserved roles of phosphates in biochemistry, provide an increasingly clear message that understanding phosphate chemistry will be a key element in elucidating the origins of life on Earth.

## 1. Introduction

Phosphates are essential to modern biological systems, and their wide and varied range of biological roles is a testament to their value in controlling chemistry and building robust structures in an aqueous environment. They provide the stable ligation required to fix information in RNA and DNA, contribute to cellular structure in phospholipids, serve as the basic currency of biochemical energy (e.g., **ATP**, phosphoenol pyruvate (**PEP**), creatine phosphate (**CP**)) (Figure 1), and feature in a wide variety of metabolites and commonly observed post-translational protein modifications [1].

Westheimer provided a detailed analysis of the essential role of phosphate in living systems more than 30 years ago [2], highlighting in particular, that phosphates are ionized at physiological pH, due to a low first pK_a_ (pK_a_ = 2.2, 7.2, 12.3). This ionic character renders phosphates hydrophilic, and facilitates their retention within a cell membrane. Importantly, in the case of RNA and DNA ligation, the ionic structure of phosphates allows the ligation of two nucleosides whilst retaining a negative charge at the phosphodiester. The charge carried by the phosphodiester ligations between nucleotides provides an essential solubilising element, and importantly, protects the phosphodiesters from hydrolysis. Beyond the structural role phosphates play in biology, they also serve a multitude of integral roles in energy metabolism, where again the kinetic stability afforded by ionization is an essential element in exploiting phosphates. Kinetic stability and thermodynamic activation are coupled to excellent effect in phosphate moieties, to provide a robust chemical drive for biochemical transformations, whilst allowing enzymatic modulation of reactions, and regulation of metabolic pathways [3].

Given the deep-seated role phosphate plays in life’s most highly conserved processes, it is essential to consider the role of phosphate at the origins of life. Numerous proposals have been made that the earliest stages of evolution might not have used a phosphorus-based biochemistry, including phosphate-free metabolism [4], lipids [5,6,7,8], and genetics [9,10,11,12], however, these hypotheses do not address the fundamental question of how biology became addicted to phosphate; they merely postpone this question. Therefore, we will focus here on the chemistry of phosphate, and specifically P(V) phosphorus chemistry, which is the cornerstone of extant metabolism. We do not aim to review all phosphate chemistry relevant to the origins of life; rather, we seek to highlight a few key elements of phosphate chemistry that we have found especially instructive in our investigations of prebiotic chemistry.

At this juncture, it must be noted that the availability of inorganic phosphate on the early Earth has been widely debated. Orthophosphate (PO_4_^3−^), the most common form of phosphorus on the Earth [13], is largely found as apatite minerals, which are relatively insoluble in most geological settings. The insolubility of apatite minerals has been referred to as the ‘phosphate problem’ [14,15,16], and of course, useful production of soluble phosphate requires that this phosphate remains soluble long enough, and in sufficient quantity, to be utilised in prebiotic chemistry. This may imply a specific environment is required for exploitation of phosphate. For example, one might consider an aqueous environment low in soluble calcium, which would be expected to rapidly precipitate orthophosphate, might be essential to exploit phosphate at the origins of life, and that the key role of phosphates in life (and prebiotic chemistry) suggests that the liberation of phosphate by other bulk anions may have played a key role in the origins of life. The challenges imposed by geochemically accessing phosphorus in the environment are certainly not trivial, but these constraints suggest it is likely that phosphate (and its availability) could play a pivotal role in bringing together prebiotic chemistry and early Earth geochemical models, as well as a pivotal role in the origins of life itself. Several reviews have discussed the availability of phosphorus on the early Earth, and numerous attempts to quantify phosphorus availability on the primitive Earth have been made (Scheme 1) [17,18,19,20,21,22,23]. We do not intend to recover this ground here; whilst the issue of phosphate availability will continue to stimulate debate within the field of prebiotic chemistry, it is clear that multiple mechanisms for the accumulation of useful phosphates under specific conditions on the early Earth are at least plausible. Indeed, the total amount of soluble phosphate in the entire hydrosphere need not be high if prebiotic phosphorus chemistry was confined to specific (phosphorus-rich) niches [16,17,24]. Accordingly, effective and selective mechanisms for phosphate incorporation into organic molecules may pose a greater challenge to prebiotic chemistry than the global phosphorus inventory, and here, we will focus upon illustrative examples of phosphate chemistry in prebiotic reactions.

## 2. Phosphorylation in Water

Electrophilic phosphorylation reactions in water require a high degree of selectivity, for example, the phosphorylation of a (nucleotide) hydroxyl moiety in water requires direct competition with solvent (55 M water), which is (weakly) nucleophilic, and can be phosphorylated to yield inorganic phosphate. The most widely investigated approach to prebiotic phosphorylation has been by electrophilic activation of orthophosphate to generate activated (anhydride-type) intermediates, which can in turn react with nucleophiles (e.g., hydroxyl groups) to afford phosphorylated products (e.g., phosphate esters). numerous activating agents have been investigated, and cyanoacetylene (**1**) [25], cyanogen (**2**) [26], cyanamide (**3**) [27], and cyanate (**4**) [28], are all noted worthy examples of prebiotically plausible electrophiles that have been exploited in this role. However, aqueous electrophilic phosphorylation reactions can suffer dramatically from adverse competition with water, and typically aqueous phosphorylations are low yielding. For example, phosphorylation of 0.16 M uridine (**5**) with cyanoformamide (**6**) in pH 8 1 M phosphate solution furnishes only 1–4% uridine-5’-phosphate (**UMP**) (Scheme 2) [25,28]. Incubation of sugars (for example, ribose (**7**)) with cyanogen (**2**) or cyanamide (**3**) in phosphate solution, also affords phosphorylated sugars (for example, ribose-1-phosphate (**R1P**)), but again with low yield (10–20%) [26,27].

The soluble phosphates obtained by the oxidation of schreibersite, have been considered for localised delivery of phosphate [29], and schreibersite has been used as a direct phosphorus source for electrophilic phosphorylation. For example, aqueous solutions of glycerol and schreibersite incubated at 65 °C under anaerobic conditions afforded glycerol phosphate (2.5%) [30], and minerals related to schreibersite, such as Fe_3_P and Fe_2_NiP, when combined with nucleosides in aqueous solution, also return mixtures of nucleotides in low yield (up to 1–6% of the total dissolved phosphorus) [31]. However, model studies of the aqueous oxidation of Fe_3_P (used as a model for schreibersite), affords significant quantities of orthophosphate and pyrophosphate, alongside minor reduced phosphorus species [20,21,22]. The amount of condensed phosphates returned have been increased by hydrogen peroxide oxidation of the reduced-phosphorus species corroded from schreibersite [23], to afford a good yield (up to 34%) of condensed phosphates, including pyrophosphate, triphosphate, and cyclotriphosphate (**cTMP**); the rest of the phosphorus is returned as orthophosphate.

## 3. Phosphorylation in Water with Polyphosphates

Polyphosphates play a dominant role in biological activation and phosphorylation; they are also intrinsically activated and generally more soluble than orthophosphate. Consequently, phosphorylation reactions exploiting condensed phosphates have drawn significant attention. Numerous attempts have been made to convert orthophosphate to polyphosphates, and pyrophosphate and triphosphate synthesis has been achieved by various condensing agents in water [32,33,34,35], but perhaps the simplest conditions for polyphosphate synthesis are heating phosphates (H_3_PO_4_, NaH_2_PO·H_2_O, NH_4_H_2_PO_4_, etc.) in the dry-state at high temperature (80–160 °C). Dry-heating orthophosphate salts furnishes pyrophosphate (5–50%), and triphosphate (1–30%), in 2 h at 160 °C [36], and even at low temperature (37 °C), insoluble calcium and magnesium orthophosphates have been condensed to afford pyrophosphate (albeit in 0.06% yield) [37]. Condensed phosphates have also been produced by magmatic processing of mixtures of apatite and basalt [38], and the simplicity and robustness of polyphosphate synthesis strongly suggests polyphosphates may have played an important role in the origin of life, as well as extant biology. Moreover, urea (**8**), a simple, highly prebiotically plausible compound, has been demonstrated to significantly improve phosphate polymerisation. Orthophosphate has been converted to polyphosphates in high yield (60%) at moderate temperatures (72 °C in 26 days) in the presence of urea (**8**), and under similar conditions, in the presence of nucleosides, cyclotrimetaphosphate (**cTMP**) has been obtained in 23% yield [39]. The most facile transformations have been observed with NH_4_H_2_PO_4_, and the related mineral struvite (MgNH_4_PO_4_·6H_2_O), which can be formed from soluble phosphates when ammonium concentrations exceed 0.01 M [40], and appears to be kinetically favoured over the more thermodynamically stable hydroxyapatite [41]. Struvite has been exploited in the phosphorylation of nucleosides [40], and pyrophosphate can be obtained from struvite in up to 88% after 48 h at 85 °C [42].

Linear polyphosphates, such as sodium triphosphate, and adenosine (**9**), afford mixtures of adenosine 2’, 3’-, and 5’-phosphates in low yield (1%) when they are refluxed in basic aqueous solutions for short periods (4–6 h). Though pyrophosphate is not observed to afford nucleotides under similar conditions, longer polyphosphates, such as cyclic hexametaphosphate, afford the same set of products as triphosphate, and in comparable yield [43]. However, of the various polyphosphates examined, cyclotrimetaphosphate (**cTMP**) is of particular note (Scheme 3). Incubation of sodium **cTMP** with **9** at high pH affords a mixture of 2’- and 3’-phosphates (**9**-2p and **9**-3p; ~1:1) in 31% yield. It is also of note that phosphorylation of 2’-deoxyadenosine (**10**), is an order of magnitude less efficient under similar conditions, and only affords a mixture of 5′- and 3’-phosphates in 2% yield [44]. Moreover, phosphorylation of adenosine-5’-phosphate (**AMP**) with **cTMP** at 100 °C yields only small quantities of nucleotide polyphosphates (**ADP** and **ATP** in 0.03% and 0.09% yield, respectively), accompanied by significant decomposition to adenosine (**9**) and free base (adenine) [45]. These results make clear the importance of the vicinal diol in efficient **cTMP**-phosphorylation of ribonucleotides. The **cTMP**-mediated phosphorylation of **9** can also be performed at neutral pH, if Mg^2+^ is present, however the reaction is sluggish and the absolute selectivity for diol phosphorylation is lost, affording a mixture of 2′,3′-cyclicphosphate **11** (3.8%) and **AMP** (<1%) [46]. Interestingly, however, exploiting a mixture of tetramethylammonium and sodium counter ions with **cTMP** results in drastically improved reactivity, and furnishes a mixture of 2’- and 3’-phosphates in 70–90% yield (at high pH) [47]. Nucleotide-5’-phoshates were not detected in these experiments, demonstrating the remarkable selectivity for diol-phosphorylation in water. The suspected mechanism for these high yielding reactions involves specific base-catalysed phosphorylation of vicinal diol by **cTMP** to yield the nucleotide 2’- or 3’-triphosphate. The significantly lowered pK_a_ of the diol results in a highly regioselective triphosphorylation, and the close proximity of the second alcohol moiety of the diol then results in attack at the α-phosphorus to form a 2’,3’-cyclic phosphate, whilst liberating pyrophosphate. The strained cyclic phosphate then undergoes alkaline hydrolysis to afford a mixture of **9**-2p and **9**-3p in excellent yield (Scheme 3). Though nucleotide 5’-phosphates are universally exploited in extant biology, it is of note that RNA hydrolysis proceeds via the transient formation of 2’,3’-cyclic phosphates. Therefore, it also appears likely that prebiotic RNA recycling would exploit 2’,3’-cyclic phosphates, and 2’,3’-cyclic phosphates would provide the most direct entry point for the advent of continuous (prebiotic) RNA evolution strategies. Though these **cTMP**-mediated phosphorylations require preformed ribonucleoside, they are highly instructive for understanding the value of 2’,3’-phosphates during the stepwise synthesis of ribonucleotides, and we will return to the chemistry of nucleotide-2’,3’-phosphates in due course.

Phosphorylations with **cTMP** have found many applications beyond prebiotic nucleotide synthesis. For example, small (simple) oligopeptides can be readily obtained by **cTMP**-mediated coupling of amino acids in aqueous solution [48,49,50]. Incubation of glycine (Gly) with **cTMP** affords diglycine in 35% yield at 70 °C after 70 h (Scheme 4); even at room temperature, the dipeptide is obtained in moderate yield (20–22%) after 190 h. Alanine (Ala) and aspartic (Asp) acid have also been observed to afford their corresponding dipeptides, however, the yields (12% and 2–5%, respectively) [48,49] are significantly suppressed relative to diglycine [50]. Serine (Ser), on the other hand, has only been observed to undergo *O*-phosphorylation, to afford small amounts of *O*-phosphoserine (**12**; 4%) as the only product [49,51].

The proposed mechanism for **cTMP** activation of amino acids occurs by a two-step mechanism (Scheme 5). First, nucleophilic attack of the amine moiety at phosphorus results in the formation of an *N*-triphosphoramidate **13**, then formation of a five-membered cyclic mixed anhydride intermediate (phosphoramidate **14**) occurs by attack of the carboxylate on the same phosphorus, coupled to displacement of pyrophosphate. Nucleophilic attack at carbon of the (activated) mixed anhydride **14**, by the amine moiety of another amino acid, then affords a *N*-phosphoro-dipeptide and dipeptide upon phosphoramide hydrolysis [52,53]. It is of note that this mechanism proceeds via a cyclic intermediate akin to the phosphorylation of the vicinal diol of a nucleotide.

The mechanism for **cTMP**-mediated peptide activation specifically favours the activation of (simple proteinogenic) amino acids, such as glycine, whereas *β*-amino acids, such as *β*-alanine **15**, that are widely viewed to be prebiotic, but not found in the proteome, are not observed to be ligated by **cTMP**. Even incubation of *β*-alanine *N*-triphosphoramidate (**16**) does not result in activation [54,55], but upon acidification, cyclisation of the amidotriphosphate occurs, to yield cyclotrimetaphosphate (**cTMP**) [56,57]. It is thought that the homologous *β*-alanine (**15**) has a higher activation energy associated with cyclisation, and the six-membered mixed anhydride is therefore kinetically prohibited [58]. This mode of α-amino acid selective ligation could be a key mechanism to select for the proteinogenic amino acids found in biology, however, it may also provide clues to why serine is ineffectual, given that the hydroxyl moiety of serine is ideally positioned to intercept the *N*-triphosphoramidate, to prevent activation by cyclic phosphoramidate formation. Moreover, though appreciable yields of diglycine (35%) can be formed by **cTMP** activation, further oligomerization to yield triglycine is only observed to occur in very low yield (<1%). A marginally improved yield of oligoglycines can be obtained by stepwise ligation; when diglycine is incubated with **cTMP** under neutral or slightly acidic conditions, moderate heating (38 °C) yields mixtures of tetraglycine and hexaglycine (15% and 4%, respectively). In contrast to the reaction with amino acid monomers, the formation a cyclic phosphoramidate is not possible; a plausible mechanism for dimer ligation involves formation of linear acyl-*O*-triphosphate **17** from the zwitterionic dipeptide, followed by dipeptide ligation. It is of note that different pH conditions are required for **cTMP**-mediated monomer and dimer ligation, and even then, dimer ligation is not efficient. The amine moiety of the dipeptide has a lower pK_a_ than the amine moiety of the monomer (8.1 vs. 9.6), therefore it is likely the formation of the *N*-triphosphate is more facile for the dipeptide than the monomer under the conditions of these experiments. Phosphorylation of the dimer does not result in cyclisation (and cannot activate the carboxylate moiety), therefore, this would account for the especially low yields of tripeptide in the reaction of glycine, diglycine, and **cTMP** at basic pH. Significant **cTMP**-phosphorylation of the dipeptide (to afford the *N*-triphosphate-glygly (**18**)) blocks further reaction with the cyclic acylphosphoramidate **14** that is required to afford trimer (Scheme 6) [59]. Although the efficient ligation of peptides with **cTMP** seems limited to glycine dimer (GlyGly), the intramolecular phosphorylation of the carboxylate group of glycine (Gly) foreshadowed the remarkable use of amine catalysis in the phosphorylation of sugars by **cTMP**.

## 4. Phosphorylation in Water with Amine Catalysis

Glyceric acid 2- and 3-monophosphate **19**-2p and **19**-3p, which are key intermediates of glycolysis, can be readily obtained in up to 40% yield in alkaline **cTMP**-mediated phosphorylation [60], or under slightly acidic conditions (pH ≥ 6.0), when facilitated by charged-mediated absorption into a mineral bilayer (Scheme 7) [61]. The reaction of glucose and sucrose with **cTMP** under alkaline (pH > 13) conditions has been observed to afford saccharide mono- and tri-phosphate derivatives at room temperature [62]. However, glycolaldehyde phosphate (**GCP**) [63,64,65], undergoes homoaldol condensation under the alkaline conditions required for phosphorylation, with **cTMP** hampering these reactions [64]. Similar problems have also been encountered for the synthesis of glyceraldehyde phosphate derivatives from glyceraldehyde (**GA**) by **cTMP**-mediated phosphorylation, and the former undergoes facile (E1cB) elimination under basic conditions. However, these problems can be readily ameliorated by amine catalysis. The reaction of ammonia with **cTMP** yields amidotriphosphate (**AmTP**) in excellent yield, and the amido group of **AmTP** is capable of reversible imine formation with carbonyl groups. The reversible capture of **AmTP** tethers the activated phosphate to the sugar substrates, and once **AmTP** is tethered to the anomeric carbon, the α-phosphate is intramolecularly delivered to the α-hydroxyl with exceptional control. This reactivity was exploited by Krishnamurthy et al. in the synthesis of glycolaldehyde phosphate (**GCP**) [64]. Further development of the strategy was undertaken by Mullen and Sutherland [66], who demonstrated that β-hydroxy-*n*-alkylamines react readily with **cTMP** to form amphiphiles. Interestingly, they observed that a hydrophobic effect dictated the product distribution of this reaction, such that short-chain β-hydroxy amines afford only phosphoramide products, but intramolecular transfer and stable amphiphilic phosphate esters are obtained from long-chain β-hydroxy amines.

Eschenmoser’s tethered α-phosphorylation strategy has been extended to the syntheses of several sugar phosphates. Glyceraldehyde (**GA**) can be converted specifically to glyceraldehyde-2-phosphate (**GA2P**), but the tetrose and pentose sugars (such as ribose (**7**)) react to afford 1,2-cyclic, and if the 2,3-hydroxyl moieties are *cis*-disposed, then also 2,3-cyclic phosphates. Therefore, the phosphorylation of ribose (**7**) proceeds initially like glyceraldehyde (**GA**), but then ‘*with a twist’,* the full activation potential of **AmTP** (the **AmTP** α-phosphate is activated twice, once with a pyrophosphate leaving group, and once with an ammonia leaving group) is utilised to give ribose-1,2-cyclic phosphate (*ribo*-**20**), and ribose-2,3-cyclic phosphate (**21**) (Scheme 7), which return the corresponding 2- or 3-phosphates upon acidification. Conversely, due to *trans*-disposition of the 2,3-diol moiety of arabinose (**22**), similar 3-phosphorylation cannot lead to the formation of a 2,3-cyclic phosphate, and therefore, only delivers arabinose-1,2-phosphate (*arabino*-**20**). Further ammonolysis of **AmTP** affords diamidophosphate (**DAP**), which phosphorylates aldoses by a similar mechanism to **AmTP**, but does not require Mg^2+^, and in the case of aldopentose sugars, higher yields are obtained with **DAP** (71% vs. 29%) [67]. The reaction of glycolaldehyde (**GC**) or glyceraldehyde (**GA**) with **DAP** furnishes the α-phosphate derivatives in excellent yield (>90%), and we have recently exploited this reaction, and these simple sugar phosphates, to develop a network of prebiotically plausible reactions that synthesise all of the intermediates of triose glycolysis [68]. For example, glyceraldehyde-2-phosphate (**GA2P**) obtained by **DAP** phosphorylation can be readily oxidised to afford glyceric acid 2-phosphate (**19**-2p), or dehydrated in pH 7 phosphate buffer to afford phosphoenol pyruvaldehyde (**23**) (Scheme 8), which can be oxidised to deliver phosphoenol pyruvate (**PEP**)—biology’s highest energy phosphate—in excellent yield, following the trajectory of triose glycolysis observed in extant biology [68].

## 5. Nucleophilic Phosphorylation

Some of the problems associated with the electrophilic phosphorylation of hydroxyl moieties can be overcome by tethering strategies, however, though these strategies have yielded excellent syntheses of simple sugar phosphates and even phosphoenol pyruvate (**PEP**) in water, most major phosphorylation targets in prebiotic chemistry are isolated hydroxyl moieties (e.g., nucleosides) which do not have suitably positioned carbonyl moieties to facilitate delivery of phosphate. Electrophilic phosphorylation strategies that employ substrate tethering to facilitate intramolecular delivery of phosphate, can rely upon very weakly activated electrophilic phosphorus centres, however, intermolecular delivery often requires higher levels of activation, making these strategies particularly susceptible to competitive hydrolysis. Accordingly, it is valuable to consider other approaches. The incredibly important role that phosphate ionization plays in the character of phosphates (at physiological pH), due its low first and second pK_a_s (pK_a_ = 2.2, 7.2, 12.3), has already been noted [2], but this ionization also opens a different perspective on phosphate chemistry at neutral pH. Whilst ionization provides kinetic stability to phosphate (di)esters, it also imparts significant nucleophilicity to phosphate/phosphate monoesters at neutral pH, where the second ionization state is readily accessed. This nucleophilicity is implicitly exploited in the electrophilic activation of phosphate, for example, through reaction of phosphate with cyanate (**4**) to accrue electrophilic activation, however, if electrophilic activation can be accrued in the substrate, then the direct phosphorylation of the substrate can be achieved through reaction with phosphate in water. This is a particularly advantageous strategy because phosphate (unlike hydroxyl groups) is significantly more nucleophilic than water at neutral pH. This is a classic method to introduce ester and phosphoester moieties in organic synthesis (e.g., Mitsonobu-type reactivity), however, at first glance, the hydroxyl motifs that are key phosphorylation targets in prebiotic chemistry, do not easily lend themselves to activation as electrophiles under prebiotic conditions. These problems are, however, a result of perspective, and can be overcome by considering the approach to activation. Rather than specifically synthesising a hydroxyl moiety and then pursuing subsequent activation of this hydroxyl group (following a classic Mitsonobu strategy), which would be very challenging under prebiotic constraints, simple activating strategies can be built into a chemical synthesis of a substrate from the start, without requiring an intermediate hydroxyl moiety.

During a seminal contribution to the investigation of prebiotic sugar phosphate synthesis, Eschenmoser and co-workers reported an exemplary nucleophilic phosphorylation in aqueous solution using orthophosphate. They observed the phosphorylation of oxarinecarbonitrile **24** to yield glycoaldehyde phosphate (**GCP**), or its cyanohydrin (**GCP**·HCN), in good yields under basic conditions (Scheme 9) [69]. Similar chemistry was used by the same group in the synthesis of phosphoserine (**12**), constitutionally related to glycoaldehyde phosphate (**GCP**), by opening of aziridine-2-carbonitrile (**25**) in moderate yield (**26**, 50% after recrystallization) in acetonitrile [70]. The prebiotic synthesis of oxirane **24** or aziridine **25** has yet to be demonstrated, but these reactions provide valuable mechanistic insights; they make use of the nucleophilicity of phosphate to achieve substrate phosphorylation, rather than relying on addition to electrophilically activated phosphate.

Recently, we have exploited a similar approach to demonstrate a prebiotic synthesis of aminooxazoline-5’-phosphates (**27**), which are key intermediates in the prebiotic synthesis of nucleotides. Previously, the synthesis of **27** had been achieved by reaction of ribose-5’-phosphate (**R5P**) and cyanamide (**3**) [71], or glyceraldehyde-3-phosphate (**G3P**) and 2-aminooxazole (**2AO**) [72]. However, neither **R5P** nor **G3P** are prebiotically accessible, moreover, under neutral and basic conditions, **G3P** undergoes facile (E1cB) elimination to give pyruvaldehyde (**28**) [73,74]. To exploit nucleophilic phosphorylation, but now in a prebiotically plausible system, we began our synthesis with acrolein (**29**) (Scheme 10). Oxidation of **29** at near neutral pH by hydrogen peroxide furnishes an excellent yield of glycidaldehyde (**30**) (>90% at pH 7.5–9). This reaction provides the oxirane moiety and activation required for phosphorylation in water. Glycidaldehyde (**30**) can then be directly phosphorylated in neutral aqueous solution by inorganic phosphate to afford **G3P**, but **G3P**, as expected, rapidly eliminates. This elimination can be readily sidestepped, however, if glycidaldehyde (**30**) first reacts with 2-aminooxazole (**2AO**), and then phosphate. The reaction of **2AO** and **30** yield a five-carbon sugar moiety that is activated (at the C5’-carbon atom) to nucleophilic substitution. The epoxide moiety of intermediate **31** can be phosphorylated by direct addition of inorganic phosphate to afford **27** in the first prebiotically plausible reaction process that specifically delivers the natural 5’-phosphorylation patterns observed in canonical nucleotides [75]. Reversing the order of aminooxazoline assembly not only yielded a highly selective 5’-phosphorylation, but also, by introducing the phosphate at a distal position from the anomeric centre, prohibits the previous deleterious E1cB elimination. The use of nucleophilic, rather than electrophilic, phosphorylation, opens a wide palette of site selective reactivity, and clearly warrants further investigation in the context of prebiotic chemistry, however nucleophilic phosphorylation strategies must be carefully considered as reactivity is necessarily orchestrated by the preceding chemical reactions, and therefore, by in-built activation accrued within an organic substrate.

## 6. Phosphorylation in Dry-States Using Condensed Phosphates

As noted above, the competing effects of water have a tendency to significantly lower the yield of phosphorylation reactions. Accordingly, excluding water through the simple process of evaporation offers a remarkably interesting scenario in which to investigate prebiotic phosphorylation. Drying is readily achieved by evaporation of water from a desired phosphorylation, and is a process that is very easily envisaged on the early Earth. These dry-state reactions are of particular interest, because they obfuscate the requirement for condensing agents, and rely only upon physical processing of phosphate rich media.

Dry-state phosphorylation using orthophosphate as the source of phosphorus has been applied to the synthesis of a wide range of molecules of prebiotic interest, however, phospholipids [76,77,78], and nucleotides [79,80,81], which both carry a phosphate moiety essential to their function/structure, have received the most attention. The phosphorylation of nucleotides, for example, proceeds rapidly at high temperatures (16% phosphorylation after 2 h at 180 °C) [79], but requires several months to achieve similar results at milder temperatures (65–85 °C). Total yields are limited by equilibration of phosphorylated and non-phosphorylated products [81], but high levels of phosphate incorporation can be driven by the irreversible formation of 2’,3’-cyclic phosphate. The efficiency of dry-state phosphorylation reactions is significantly improved by the inclusion of urea (**8**) [82]. After evaporation, urea (**8**) acts as both a catalyst for phosphoryl transfer and as a pseudo-solvent, providing fluidity at elevated temperatures. Phosphorylation of nucleosides with mixtures of urea (**8**), ammonium chloride and various phosphates (Na_2_HPO_4_ and Ca_5_(PO_4_)_3_OH) at temperatures ranging from 60 to 100 °C, afford nucleotides. High degrees of total phosphate incorporation are observed (>96% for pyrimidine nucleotides). Ammonium salts are also commonly used to avoid carbamylation (upon loss of ammonia from urea (**8**)) and assist with the acidification of the reaction mixture, which promotes phosphorus transfer [83,84,85,86].

The precise mechanistic role of urea (**8**) has not been proven, however, it is likely that urea (**8**) displaces water from a tautomeric form of monoanionic phosphate, in which the charge state facilitates dissociative loss of water, and nucleophilic attack (by urea (**8**)), to generate an activated ureidophosphate intermediate **32** capable of transferring phosphate between hydroxyl moieties (Scheme 11). Further evidence for this mechanism is found in the reversible phosphorylation of hydroxyl groups, but the irreversible synthesis of 2’,3’-cyclic phosphates. Accordingly, and necessarily, a different mechanism for cyclisation of 2’,3’-cyclic phosphates must operate under these conditions. It seems likely that due to the high effective molarity of the 2’-hydroxyl to a 3’-phosphate (or vice versa), the phosphorylation mechanism can switch to an associative mechanism, therefore allowing cyclic phosphate synthesis, but prohibiting the degradation of 2’,3’-cyclic phosphates under these conditions. It is the mechanistic switch that is responsible for the highly effective accumulation of phosphate in ribonucleotides under these conditions, and therefore potentially an important contributing factor in the selection of ribonucleotides as key metabolites to support life from prebiotic chemistry.

This dry-state urea-mediated phosphorylation protocol has been applied to the phosphorylation of various anhydronucleotides to good effect [86,87]. Upon phosphorylation of anhydronucleotides, under the conditions of urea-mediated phosphoryl-transfer, a third mechanism comes into play, exploiting the activation inherent in the anhydronucleotide bond to synthesise a cyclic phosphate; but rather than by dissociative phosphoryl-transfer, is followed by associative cyclisation, now dissociative phosphoryl-transfer is followed by intramolecular rearrangement by attack of (tethered) nucleophilic phosphate on an activated carbon atom. Akin to the activation that was discussed above, built into the synthesis of an oxirane, here, the stepwise assembly of an anhydronucleotide can be used to chemospecifically direct the activation of a carbon atom to facilitate the synthesis of a new phosphate group. As this phosphorylation mechanism exploits the activation of the C2’-carbon atom to S_N_2-nucleophilic displacement, anhydro-arabinofuranosyl nucleosides *arabino*-**33** and *arabino*-**34** are phosphorylated, and rearrange to deliver the natural ribofuranosyl nucleotides as their stable 2’,3’-cyclic phosphates **35** and **36**. The phosphorylation of anhydronucleotides displays remarkable selectivity for phosphorylation of the 3’-OH [86,87], as a result of both the kinetic (n→π* donation supresses nucleophilicity of the 5’-OH) and thermodynamic (irreversible 2’,3’-cyclic phosphate synthesis) characteristics [86,87,88]. To further emphasise the potential of this strategy, we have recently applied the urea-mediated phosphorylation to the divergent synthesis of pyrimidine and 8-oxo-purine ribonucleotide, wherein it is remarkable to note that a minor modification of the purine moiety (C8 oxidation) leads to divergent reaction pathways to canonical pyrimidine nucleotides and 8-oxo-purine ribonucleotides, from one common intermediate. Importantly, though the specific heterocyclic motifs of the intermediate anhydro-nucleotides *arabino*-**33** and *arabino*-**34** (Scheme 12) are different, they both display n→π* suppression of 5’-OH nucleophilicity, and both undergo 3’-OH selective phosphorylation, followed by intramolecular inversion to yield the natural β-*ribo*-stereochemistry.

Once again, through careful consideration of the method used to construct the organic substrates, it was possible to build two substrates (one purine and one pyrimidine) that both directed phosphorylation chemistry via the same chemical strategy. Both pathways exploit dry-state dissociative phosphoryl-transfer, that is (partially) controlled through stereoelectronic effects, both pathways exploit tethered nucleophilic attack to deliver a new cyclic phosphate moiety, and finally, both pathways exploited in-built electrophilic activation of a carbon atom through the formation of an anhydronucleotide bond to the C2’ carbon atom. Though more work is required to realise a prebiotic synthesis of the canonical purines, the continued success of these strategies warrants further investigation of dry-state nucleotide phosphorylations.

Beyond the phosphorylation of nucleotides, the reversible characteristics of urea-mediated phosphorylations have also been exploited to furnish phospholipid-type molecules selectively from a mixture of alcohols. Dry-state phosphorylation exploits evaporation to remove water from phosphorylation reactions, and accordingly, simple alcohols (such as the oxygenate products of Fischer–Tropsch synthesis) can be readily fractionated by evaporation during these phosphorylation reactions. Long chain alcohols, such as decanol (**37**), can be phosphorylated in preference to shorter chain alcohols, such as hexanol (**38**) and ethanol (**39**), to selectively yield decyl-phosphate (**40**), by simple virtue of comparative volatility (Scheme 13) [29].

The ability of urea (**8**) to facilitate phosphoryl-transfer derives from the nucleophilicity of the urea oxygen atom towards phosphorus electrophiles. This property is certainly not unique to urea (**8**), for example, formamide will also (albeit less efficiently, likely due to significantly reduced nucleophilicity of the amide vs. urea oxygen) promote phosphoryl-transfer [86,87,89,90,91,92].

The excellent efficiency and selectivity, and the remarkable simplicity of amide/urea-mediated phosphorylation, will no doubt see its continued application in unpicking the origins of life, and the sheer volume of research, and the number of years that these phosphorylation strategies have been employed, are a testament to the efficiency and reproducibility of dry-state phosphorylation as a method, and perhaps this, more than anything else, is an indication that it might come to be recognised as a key element in facilitating the incorporation of phosphate in living systems.

## 7. Phosphate as a Catalyst

As has been made clear in the above discussions, phosphate is an important constitutional component of life, and numerous methods for its incorporation into metabolites under prebiotic conditions are being developed. Another feature of prebiotic phosphate chemistry, which has received far less attention, but can be nonetheless striking in its application, are the catalytic and (pH/chemical) buffering capabilities of phosphate. Whilst phosphate is inherently required as a constitutional component for the assembly of RNA, phosphoenol pyruvate (**PEP**), phospholipids, etc., it has also been shown to have a critical, multi-faceted, and often very subtle non-structural role in prebiotic synthesis. For example, during the prebiotic synthesis of pyrimidine ribonucleotides reported by Powner et al. [87], all steps in the sequence from 2- and 3-carbon atom building blocks are carried out in the presence of phosphate (Scheme 14). The phosphate is ultimately the source of phosphorus for urea-mediated phosphorylation and 2′-inversion [87], and it was therefore deemed important to demonstrate phosphate could be present from the start, and throughout the reaction sequence, however, phosphate also performs numerous other crucial roles at critical stages in the synthesis. For example, whilst most steps proceed efficiently at near-neutral pH, the formation of 2-aminooxazole (**2AO**), from glycolaldehyde (**GC**) and cyanamide (**3**), initially appeared to require alkaline conditions, to allow the key base-catalysed steps to proceed smoothly [93]; indeed, a complex mixture of intermediate addition products were returned in neutral water. The requirement for high-pH conditions in this first step are negated, however, in the presence of phosphate buffer. A high yield of **2AO** is obtained at pH 7 in phosphate solution, due to the efficient general acid–base catalysis exhibited by the phosphate anion/dianion (pK_a_ 7.2). By lowering the pH of this first step (through general acid–base catalysis), phosphate brings this step into consonance with the rest of the route that requires near neutral pH conditions (pH 6–7). Furthermore, excess cyanamide (**3**), which has the potential to interfere in later stages, particularly upon introduction of glyceraldehyde (**GA**), slowly reacts in the presence of phosphate to afford urea (**8**), a compound that is later instrumental in catalysing the phosphorylation/rearrangement step (discussed above). Phosphate also acts as a pH buffer in the reaction of **2AO** with glyceraldehyde (**GA**) to give the aminooxazolines **41**, as a mixture of diasteriomers (*arabino*:*ribo*:*xylo*:*lyxo* 15:25:6:4). Ribose aminoxazoline (*ribo*-**41**) is the least soluble of these, and can be isolated by direct crystallization; as such, the arabinose aminooxazoline (*arabino*-**41**) becomes the major product in the supernatant [94]. The precipitated ribose aminooxazoline (*ribo*-**41**) can be redissolved in phosphate solution, and equilibrated with the *arabino*-configuration via a phosphate-mediated isomerisation process, where again, phosphate acts as a general acid–base catalyst for the epimerisation of C2’-carbon atom of the aminooxazolines **41** [95].

Phosphate plays a particularly essential role in the reaction of the aminooxazolines (**41**) with cyanoacetylene (**1**). In water, the reaction of **41** and **1** results in the formation of cytidine nucleotides [71,96]; ribose aminooxazoline (*ribo*-**41**) affords α-cytidine (α-**42**), and arabinose aminooxazoline (*arabino*-**41**) gives β-arabinosyl-cytidine (**43**). These products, α-**42** and **43**, require further stereochemical manipulation to get to the natural β-cytidine (β-**42**), which has a greatly adverse impact on the overall yield, but perhaps more importantly (without pH buffering), poor regioselectivity of cynanovinylation is observed, and a wide range of by-products result alongside α-**42** and **43**, from these reactions. However, if the cyanovinylation is performed in the presence of phosphate buffer at pH 6.5, anhydronucleoside (**33**) is furnished in near quantitative yield. The anhydronucleotide linkage of **33** is unstable without phosphate, due to the tendency for the solution pH to increase as the reaction proceeds, but hydrolysis to α-**42** or **43** is completely suppressed in the presence of phosphate buffer. The *arabino*-anhydrocytidine (*arabino*-**33**) can then be efficiently converted to β-cytidine-2’,3’-cyclic phosphate (**35**) through urea-mediated phosphorylation and C2’-stereochemical inversion. The overall efficiency of the cyanovinylation reaction is markedly improved in the presence of phosphate, by stabilising the solution pH (alkaline), hydrolysis of both anhydronucleotide **33** and cyanoacetylene (**1**) are minimised, which would otherwise limit the yield. Finally, the phosphate in this reaction acts as a chemical buffer; excess cyanoacetylene (**1**) is sequestered by phosphate to give cyanovinyl phosphate (**44**), preventing overreaction with the desired nucleotide product, and, importantly, **44** is a phosphorylating agent in its own right that goes on to react further with phosphate, to yield pyrophosphate (PPi). Pyrophosphate can be subsequently used in the anhydronucleotide phosphorylation step to afford improved yield, and reduce hydrolytic by-products. Thus, phosphate behaves not just as a reagent in prebiotic nucleotide synthesis, but also as a catalyst, a pH buffer, and a chemical buffer, removing potentially detrimental reactive species such as cyanoacetylene (**1**) and cyanamide (**3**) from solution.

The isomerisation of glyceraldehyde (**GA**) to dihydroxyacetone (**DHA**) is promoted by phosphate catalysis. This equilibrium, which strongly favours **DHA**, had been seen as an undesired reaction, with the potential to hamper the formation of aminooxazolines (**41**) en route to ribonucleotides. Though photoreduction of **DHA** affords a lipid precursor (glycerol) and valine precursor (acetone) [89], **DHA** would be detrimental to selective ribonucleotide synthesis, opening an interesting conundrum for the origins of RNA. The phosphate rich conditions that favour nucleotide assembly, catalyse the loss of **GA** (an essential nucleotide precursor) to **DHA**. In pursuit of a solution to this ‘**DHA**-problem’, we observed that incubation of **DHA** with β-mercaptanoacetaldehyde (**BMA**) and cyanamide (**3**) in phosphate buffer (pH 7), produced pure glyceraldehyde aminal (**45**) in excellent yield (>85%) (Scheme 15). Importantly, **45** is sequestered as a crystalline precipitate, and this crystallisation inverts the thermodynamic preference for **DHA** > **GA** [97,98]. In the same way, glycolaldehyde (**GC**) and 2-aminothiazole (**2AT**) produce the corresponding crystalline aminal (**46**), however, as glycolaldehyde (**GC**) is symmetric with respect to Lobry de Bruyn–van Ekenstein transformation, and it cannot equilibrate with a ketose isomer, therefore, crystallisation of **46** is significantly more rapid than crystallisation of **45** from **DHA**. The precipitation of **45** is effectively time-resolved from precipitation of **46** through phosphate catalysed Lobry de Bruyn–van Ekenstein rearrangement, because precipitation of **45** requires slow release of **GA** from a more stable ketose isomer, **DHA**. Therefore, when complex mixtures of aldoses and ketoses (up to 26 different sugars), and **2AT** (or stoichiometric **BMA** and cyanamide (**3**)) are incubated in phosphate solution, selective sequestration of firstly, glycolaldehyde (**GC**), and then **GA**, occurs. This sequestration and resolution of C2 and then C3 sugars, solely as their aldose forms, from complex mixtures, predicates the order that is required for selective assembly of ribonucleotides from complex prebiotically plausible mixtures. Once again, phosphate acts as an essential general acid–base catalyst in this sequestration and selection process, and the phosphate-facilitated isomerization of **DHA**/**GA** is essential to the separation of these two important aldoses for the synthesis of ribonucleotides [98].

As well as catalysing aldo–keto isomerisation (Lobry de Bruyn–van Ekenstein transformation), phosphate has been shown to promote imine–enamine–aldehyde tautomerization, and the Amadori rearrangement. For example, glycolaldehyde (**GC**) can be coupled to the purine precursor 5-amino-imidazole-4-carboxamide (**47**), via an amine linkage through Amadori rearrangement by mild heating (60 °C) in phosphate solution to furnish azepinomycin (**48**)—a guanine deaminase inhibitor—in one protecting-group-free step, from low cost commercial materials (Scheme 16). This Amadori rearrangement raises the prospect of an alternative, mild method for coupling sugars with nitrogenous bases [99], but also highlights an important role that efficient prebiotic chemistry can play in delivering synthetically valuable reaction strategies that can find general application in organic chemistry. Recently, Szostak and co-workers have demonstrated the remarkable efficiency of NMP activation by 2-aminoimidazole (**2AI**) [100], which is of particular note, due to its generational relationship with 2-aminoxazole (**2AO**) and 2-aminothiazole (**2AT**) (that are discussed above) through a phosphate catalysed Amadori rearrangement (Scheme 16) [98,101].

Recently, the prebiotic synthesis of high-energy (glycolysis) metabolite phosphoenol pyruvate (**PEP**) was demonstrated [68], and once again, the synthesis highlighted the multiple roles that phosphorus can occupy in robust prebiotic synthesis. The α-phosphorylation of glycolaldehyde (**GC**) and glyceraldehyde (**GA**) by diamidophosphate (**DAP**) was rapidly and efficiently achieved in phosphate solution, where phosphate’s buffering capacity acts to off-set the detrimental pH increase due to release of ammonia, which had otherwise been found to inhibit phosphorylation, and required manual and continuous pH adjustments to offset this effect. Incubation of glyceraldehyde-2-phosphate (**GA2P**) in neutral phosphate solution then resulted in a high-yielding dehydration to furnish phosphoenol pyruvaldehyde (**23**), the direct precursor (via oxidation) to **PEP** (Scheme 8). Similarly, phosphate catalysis was demonstrated to efficiently convert glycolaldehyde phosphate (**GCP**) directly to phosphoenol pyruvaldehyde (**23**), through a one-pot aldol condensation and elimination with formaldehyde (**49**). Formation of glyceraldehyde phosphate (**GA2P**), from the aldol reaction of **GCP** and **49**, was known to occur slowly under alkaline conditions [63], but incubation at 60 °C in phosphate solution allowed a phosphate catalysed aldolisation and subsequent dehydration to be carried out rapidly and cleanly at neutral pH, once again demonstrating the remarkable effect of phosphate catalysis. The efficiency of phosphate catalysis, and the effective role that it can play catalysing proton transfer at near neutral pH in aqueous solution, suggests that there is a great deal more to discover with respect to the subtle effects that phosphate can induce within the context of prebiotic chemistry. One is struck by the simplicity of this catalyst and this simplicity, as well as its ideal pKa match for neutral pH reactivity, which is likely, in part, responsible for its broad scope of application. New roles for phosphates will continue to be found, such as the remarkable hydrotropic properties of **ATP** that lead to enhanced protein solubility [102], and it is the combination and breadth of the various attributes of phosphate chemistry that define the value of phosphates in biology and biochemistry, but when these remarkable properties are considered with the deep-seated evolutionary history of phosphate in life, it becomes more pressing than ever to respond to the availability of phosphates in a positive light, rather than retreating from this challenge.

## 8. Conclusions and Outlook

Significant progress has been made towards understanding the prebiotic chemistry of phosphorus, but is clear that the ‘phosphate problem’ is still to be conclusively resolved. However, efforts to demonstrate further prebiotic chemistry of phosphates should not be hindered by this lack of geophysical certainty; modern biochemical evidence suggests that phosphates are very likely to have played a significant part in the chemical origins of life. Moreover, there are multiple potential solutions to the problem of phosphate availability, and several plausible phosphate-producing environments can now be considered. Volcanic activity may provide ortho-, pyro-, and tripolyphosphate from apatite and basalt [38]; the same compounds can be obtained by the oxidation of schreibersite [20,21,22,23], a mineral found in meteorites. Furthermore, instead of focusing on ‘global’ phosphorus availability (in oceans), the essential role that phosphates play in biology may suggest that accumulation in local environments (e.g., ponds, lakes, or craters) played an essential role in defining the setting for prebiotic (phosphate) chemistry. In such environments, systems tailored to phosphate release and accumulation, such as cation/anion fractionation, can be readily envisaged. If these ponds, lakes, or pools are linked together by rivers or stream systems, and/or to hydrothermal activity and volcanic out-gassing, the in-flow of solutions containing prebiotic feedstocks, in combination with atmospheric delivery, may provide the nutrients required under the conditions essential to orchestrate the first steps towards life. The important role that phosphates can play in this chemistry is clear, but it is also clear that phosphate availability is highly dependent upon the given environment, therefore, as well as phosphate taking an essential role in orchestrating the initial steps of life, phosphate, by virtue of its environment dependent restrictions, could play an essential role in identifying likely scenarios and environmental constraints for the origins of life. Even the most optimistic scenarios currently suggest limited amounts of phosphorus would have been soluble in the prebiotic ocean, and this may suggest that the early oceans were not the nurseries of life, rather, local environments and conditions (for example, lake or river environments) could have yielded greater solubility of phosphorus species, through stepwise precipitation or salt leaching, and might have provided key environments for the origins of life on Earth. The unknown availability of phosphates under prebiotic constraints inevitably leads to questions over their prebiotic relevance, but their biological importance is irrefutable, moreover, the simplicity of fixing ammonia (which aids in phosphate mobilisation) through cyanide reduction, together with the valuable catalytic role that can be played by simple organic amides, such as urea (**8**) and formamide, suggests that further investigation of the availability of phosphate needs to be considered from a systems chemistry perspective [103], together with the geochemical constraints for phosphate availability. By proactively seeking environmental conditions to access phosphate, we will undoubtedly learn more about the location of life’s origin.

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
