# Peer review of "A Chemist’s Perspective on the Role of Phosphorus at the Origins of Life"

_life, 2017, doi:10.3390/life7030031_

Round 1

Reviewer 1 Report

Phosphorylations, especially in aqueous phase, are important to the emergence of life. Therefore this review is of fundamental interest to the reader of origin of life. Regarding to the outlooks, in addition to the speculation of the availability of phosphate on early Earth, there is much richer phosphate on Mars than on Earth (Adcock, C. T. et al. Nature Geosci. 2013, 6, 824.) should be addressed, since it is a vital issue for the astrobiology life searching.

Regarding to the text, the author has used extensive examples to illustrate the importance of prebiotic phosphate chemistry. But we would like to ask the author to include some contributions from China in this review.

-> Page 3

Even though typically aqueous phosphorylations are low yielding, we suggest that some relevant reactions show elegant yields in alkaline condition. Ref: Green Chem., 2009, 11, 569–573.

-> Page 5

The Phosphorylation of adenosine with trimetaphosphate under simulated prebiotic condition’ should be mentioned here. It could be a possible pathway to the formation of ATP on early earth. Ref: Origins of Life and Evolution of the Biosphere, 2002, 32, 219-224.

-> Page 7

The mechanism for amino acid activation and peptide ligation is dependable. We suggest some detailed datum about this amino-activation step. Ref: Amino Acids, 2008, 34, 47-53.

Finally, A review of N-phosphoryl amino acid models for P–N bonds in prebiotic chemical evolution mentioned the important role of N-phosphoryl amino acids in origin of life. It is worthy to mention that these compounds could be efficiently produced in the reaction of amino acids with trimetaphosphate. Ref: Science China Chemistry, 2015, 58, 374-382

Author Response

We would like to thank the reviewer for the time they have taken reviewing our manuscript.

Comment: there is much richer phosphate on Mars than on Earth (Adcock, C. T. et al. Nature Geosci. 2013, 6, 824.) should be addressed, since it is a vital issue for the astrobiology life searching. 

Response:  The relevance of Martian environments to the origins of Life on Earth is beyond the scope of our manuscript.

Comment: Even though typically aqueous phosphorylations are low yielding, we suggest that some relevant reactions show elegant yields in alkaline condition. Ref: Green Chem., 2009, 11, 569–573. 

 Response:  Though this paper is interesting in the context of developing green chemistry (using known prebiotic chemistry as a platform), we are not reviewing synthetic green chemistry and therefore this work is outside the scope of our paper.

Comment: The Phosphorylation of adenosine with trimetaphosphate under simulated prebiotic condition’ should be mentioned here. It could be a possible pathway to the formation of ATP on early earth. Ref: Origins of Life and Evolution of the Biosphere, 2002, 32, 219-224.

Response: We have already commented on the phosphorylation of adenosine, deoxyadenosine and AMP with cTMP in the manuscript. The phosphorylation yields in this reference are lower than the earlier work by Saffhill or Yamagata that we have already cited. Moreover, this section of the review (page 5) is specifically looking at phosphorylation in water, and the reference suggested predominately relates to solid phase and wet-dry cycles that are not relevant to this section.

Comment: The mechanism for amino acid activation and peptide ligation is dependable. We suggest some detailed datum about this amino-activation step. Ref: Amino Acids, 2008, 34, 47-53.

Response: We have added this citation to our manuscript.

Comment: Finally, A review of N-phosphoryl amino acid models for P–N bonds in prebiotic chemical evolution mentioned the important role of N-phosphoryl amino acids in origin of life. It is worthy to mention that these compounds could be efficiently produced in the reaction of amino acids with trimetaphosphate. Ref: Science China Chemistry, 2015, 58, 374-382

Response: We have already reviewed and provide primary references for the formation of P-N bond by the reaction of amino acids with cTMP.

Reviewer 2 Report

The present manuscript “A Chemist’s Perspective on the Role of Phosphorus at the Origins of Life” focuses the integration of phosphorus into prebiotic compounds and the role of phosphate in catalyzing other important prebiotic reactions. This review study is original, very informative and has long been awaited. It brings together geology/geochemistry and biochemistry/biology in a unique and elegant manner. I personally liked the numerous illustrative examples made by the authors, the excellent blend between old and new references, and the clear and engaging writing. In addition, I fact-checked most of the references and they were all accurate. But if I were to add a critic, I would have just wished that the phosphorylation in water section could be divided into subheadings. It was at times lengthy to me especially since it deals with different phosphate sources (orthophosphate, phosphate minerals, and polyphosphates), different substrates (amino acids, nucleosides) and with their respective detailed mechanisms of phosphorylation. The authors can also cite Costanzo et al. JBC (2007) and Gull et al. this issue, to expand the phosphorylating mineral inventory. Also, one may argue that the cTMP mediated peptide oligomerization is not a sensu stricto phosphorylation, as much as it belongs to the Phosphate as a catalyst section. Finally, to truly close the gap between geology and biology, I would have mentioned that ribozyme catalysis also undergoes similar chemistry, where the 2’OH attacks the adjacent phosphodiester bond via a SN2 type reaction. But, all these suggestions are not indispensable and the manuscript, which was a pleasure to read, can be published immediately, after minor proofreading.

Minor comments:

- Line 41: “This ionic character renders phosphates hydrophilic and facilitates their retention within a cell membrane”. I know this statement comes from the reference, but I always thought that, to be retained in a membrane, a molecule exploits its hydrophobic part rather than its hydrophilic part (see the Szostak work).

- Line 32: been produced

-  Scheme 9. Nucleophilic (missing an o) phosphorylation in water. But the second reaction is in MeCN…

Author Response

Comment: This review study is original, very informative and has long been awaited. It brings together geology/geochemistry and biochemistry/biology in a unique and elegant manner. I personally liked the numerous illustrative examples made by the authors, the excellent blend between old and new references, and the clear and engaging writing. In addition, I fact-checked most of the references and they were all accurate.

Response: We thank the reviewer for these comments.

Comment: But if I were to add a critic, I would have just wished that the phosphorylation in water section could be divided into subheadings. It was at times lengthy to me especially since it deals with different phosphate sources (orthophosphate, phosphate minerals, and polyphosphates), different substrates (amino acids, nucleosides) and with their respective detailed mechanisms of phosphorylation.

Response: We have added further subheadings to this section.

Comment: The authors can also cite Costanzo et al. JBC (2007) and Gull et al. this issue, to expand the phosphorylating mineral inventory.

Response: Though highly interesting, the inventory of mineral phosphate sources on the early Earth are beyond the scope of this manuscript. However, we would note that we have already cited Gull et al twice.

Comment: Also, one may argue that the cTMP mediated peptide oligomerization is not a sensu stricto phosphorylation, as much as it belongs to the Phosphate as a catalyst section.

Response: We resolutely believe this is the correct section. cTMP is a reagent in these reactions not a catalyst; cTMP is consumed in the reaction. The intermediated is a phosphorylated amino acid, so the first step is indeed a phosphorylation reaction. cTMP is not a catalyst in these reactions, the cTMP is not regenerated.

Comment: Finally, to truly close the gap between geology and biology, I would have mentioned that ribozyme catalysis also undergoes similar chemistry, where the 2’OH attacks the adjacent phosphodiester bond via a SN2 type reaction.

Response: Ribozymes and RNA template reactions, though again highly interesting, are beyond the scope of this manuscript. 

Comment: But, all these suggestions are not indispensable and the manuscript, which was a pleasure to read, can be published immediately, after minor proofreading.

Response: We thank the review for these comments.

Minor comments:

Comment: Line 41: “This ionic character renders phosphates hydrophilic and facilitates their retention within a cell membrane”. I know this statement comes from the reference, but I always thought that, to be retained in a membrane, a molecule exploits its hydrophobic part rather than its hydrophilic part (see the Szostak work).

Response: Membrane permeability is complex, and to my knowledge still not fully understood. However, membranes are excellent barriers to charged molecule, such as phosphates. According to Szostak et al “Contemporary phospholipid-based cell membranes are formidable barriers to the uptake of polar and charged molecules ranging from metal ions to complex nutrients …” and “local membrane deformations are required for solute passage across the membrane”.  Moreover, Szostak et al “measured nucleotide permeation by encapsulating nucleotides within vesicles and then determining the fraction of the encapsulated nucleotide that had leaked out of the vesicles at various times. Because charge has such a dominant effect in restricting solute permeation through membranes, we first examined the effect of nucleotide charge on permeation through myristoleic acid:GMM (2:1) membranes … these molecules were either too large or too highly charged to cross the membrane.”

Szostak et al: Template-directed synthesis of a genetic polymer in a model protocell Nature, 454, 122 (2008) doi:10.1038/nature07018

Comment: Line 32: been produced

Response: We did not understand the requested change?

Comment: Scheme 9. Nucleophilic (missing an o) phosphorylation in water. But the second reaction is in MeCN…

Response: Typo corrected.

Reviewer 3 Report

The review by Coggins et al. comprises a broad view on the role of phosphorus in prebiotic chemistry.  The paper deals both with plausible prebiotic formation of important biochemicals containing phosphate group and with effects that phosphorus compounds can have on prebiotic reactions in which the element does not enter into synthesized molecules.  The article is written in an excellent English, and the references are adequate to a major extent.

One aspect I found lacking. The authors write:

“Numerous proposals have been made that the earliest stages of evolution might not have used a phosphorus-based biochemistry, including phosphate-free metabolism,[4] lipids,[5–8] and genetics,[9–12] however these hypotheses don’t address the fundamental question of how biology became addicted to phosphate, they merely postpone this question.”

The authors are correct in their statement but it would be prudent to mention in such an introduction that they are indeed many works which postulate or assume existence of an early phosphate source in their considerations about origin of life. For example Piast and Wieczorek, 2017, Astrobiology 17, 277 see the emergence of phosphate transfer catalyst as crucial for origin of life, similarly Budin and Szostak 2011, PNAS, 108, 5249 believe that emergence of such catalyst would have clear evolutionary advantage. Pasek and Kee groups have made many arguments for the prebiotic presence of various phosphorus sources, ex. Pasek et al. 2008, Angewandte, 47, 7918; Bryant et al. 2013, Geochemica Cosmochim Acta, 109, 90.

All in all I would like to urge the editors of Life to publish the paper after this small correction.

Author Response

Comment: The review by Coggins et al. comprises a broad view on the role of phosphorus in prebiotic chemistry.  The paper deals both with plausible prebiotic formation of important biochemicals containing phosphate group and with effects that phosphorus compounds can have on prebiotic reactions in which the element does not enter into synthesized molecules.  The article is written in an excellent English, and the references are adequate to a major extent.

Response: We thank the reviewer for these comments.

Comment: ...it would be prudent to mention in such an introduction that they are indeed many works which postulate or assume existence of an early phosphate source in their considerations about origin of life. For example Piast and Wieczorek, 2017, Astrobiology 17, 277 see the emergence of phosphate transfer catalyst as crucial for origin of life, similarly Budin and Szostak 2011, PNAS, 108, 5249 believe that emergence of such catalyst would have clear evolutionary advantage.

Response: We have specifically reviewed informative experimental phosphorus(V) chemistries, and do not intend to provide an overview of hypothetical prebiotic uses for phosphorylation. Though both Piast/Wieczorek and Budin/Szostak provide hypothetic context for the value of a catalyst to transfer phosphate, neither resolves how this might be chemically achieve in practice, therefore, though interesting, these papers are both outside the scope of our manuscript.

Comment: Pasek and Kee groups have made many arguments for the prebiotic presence of various phosphorus sources, ex. Pasek et al. 2008, Angewandte, 47, 7918; Bryant et al. 2013, Geochemica Cosmochim Acta, 109, 90

Response: Our manuscript specifically focuses upon the biological relevant P(V) oxidation state of phosphorus, as noted in the introduction. However, we have already cited four Pasek/Kee papers, including:

Pasek, M. A.; Kee, T. P.; Bryant, D. E.; Pavlov, A. A.; Lunine, J. I. Production of Potentially Prebiotic Condensed Phosphates by Phosphorus Redox Chemistry. Angew. Chem., Inter. Ed. 2008, 47, 7918–7920, doi:10.1002/anie.200802145

All in all I would like to urge the editors of Life to publish the paper after this small correction.